# Ribosome recycling is not critical for translational coupling in *Escherichia coli*

**Kazuki Saito[1], Rachel Green[1,2], Allen R Buskirk[1]\***

[1]Department of Molecular Biology and Genetics, Baltimore, United States; [2]Howard Hughes Medical Institute, Johns Hopkins University School of Medicine, Baltimore, United States

**Abstract** We used ribosome profiling to characterize the biological role of ribosome recycling factor (RRF) in *Escherichia coli*. As expected, RRF depletion leads to enrichment of post-termination 70S complexes in 3′-UTRs. We also observe that elongating ribosomes are unable to complete translation because they are blocked by non-recycled ribosomes at stop codons. Previous studies have suggested a role for recycling in translational coupling within operons; if a ribosome remains bound to an mRNA after termination, it may re-initiate downstream. We found, however, that RRF depletion did not significantly affect coupling efficiency in reporter assays or in ribosome density genome-wide. These findings argue that re-initiation is not a major mechanism of translational coupling in *E. coli*. Finally, RRF depletion has dramatic effects on the activity of ribosome rescue factors tmRNA and ArfA. Our results provide a global view of the effects of the loss of ribosome recycling on protein synthesis in *E. coli*.

**\*For correspondence:**
buskirk@jhmi.edu

## Introduction

After the synthesis of a protein is complete, the ribosomal subunits are separated from each other and from mRNA to be reused in the next round of translation, a process known as ribosome recycling (*Janosi et al., 1996*). Although this process always involves the active dissociation of post-termination complexes (post-TCs), the molecular mechanism of recycling differs among the three domains of life (*Youngman et al., 2008*; *Buskirk and Green, 2017*). These differences are already evident in the termination step: in eukaryotes and archaea, termination is carried out by a complex containing both a release factor and a translational GTPase (eRF1 and eRF3 in eukaryotes) (*Frolova et al., 1994*; *Alkalaeva et al., 2006*; *Saito et al., 2010*). eRF1 remains in the ribosome after the release of the nascent peptide and helps recruit the factor that catalyzes subunit splitting, Rli1 (in yeast) or ABCE1 (in mammals) (*Pisarev et al., 2010*; *Shoemaker and Green, 2011*). The tRNA and small subunit are then released from the mRNA by 40S recycling factors (*Skabkin et al., 2010*). In contrast, although the bacterial release factors RF1 and RF2 share similar names with their eukaryotic counterparts, they are evolutionarily unrelated and act alone to release the nascent peptide (*Scolnick et al., 1968*). Removal of these factors by the translational GTPase RF3 clears the way for binding of ribosome recycling factor (RRF) (*Freistroffer et al., 1997*; *Peske et al., 2014*; *Koutmou et al., 2014*). In a mechanism unique to bacteria, RRF works together with the GTPase EFG to promote subunit splitting and release of the large subunit (*Janosi et al., 1996*). Binding of IF3 then excludes the deacylated tRNA from the 30S subunit and prevents reassembly of the 70S complex (*Prabhakar et al., 2017*).

Although the molecular mechanism of ribosome recycling has been worked out in great detail through biochemical and biophysical experiments for both bacteria (*Hirokawa et al., 2005*; *Borg et al., 2016*; *Prabhakar et al., 2017*) and eukaryotes (*Pisarev et al., 2010*; *Shoemaker and Green, 2011*), the physiological role of ribosome recycling has been difficult to study at the global level in vivo. Recent studies in *Saccharomyces cerevisiae* using ribosome profiling to study recycling

factors in vivo revealed that depletion of the 80S recycling factor Rli1 results in an accumulation of ribosome density at stop codons, consistent with a build-up of post-TCs that fail to be recycled, and an associated queue of elongating ribosomes that collide with these post-TCs at defined distances upstream of stop codons (*Young et al., 2015*). These studies also reported abundant ribosome density in the 3′-UTR upon Rli1 knockdown. Some of this density can be attributed to translating ribosomes that re-initiated downstream of the stop codon, although the mechanism of this re-initiation is yet to be elucidated. In a later study, Guydosh and co-workers depleted the 40S recycling factors Tma64/Tma22/Tma20, again observing stacked ribosomes upstream of the stop codon and re-initiation arising from 40S scanning ribosomes in the 3′-UTR (*Young et al., 2018*). These reports validate previous biochemical work on these ribosome recycling factors in yeast and reveal that the direct consequence of loss of recycling is unintended re-initiation in untranslated regions.

A similar ribosome profiling analysis in bacteria has been lacking, but the link between recycling and re-initiation is clearly of interest given the role re-initiation has been proposed to play in the translational coupling of genes in polycistronic transcripts (*Schümperli et al., 1982*; *Baughman and Nomura, 1983*; *Aksoy et al., 1984*; *Petersen, 1989*; *Chiaruttini et al., 1996*; *Heurgué-Hamard et al., 2002*; *Levin-Karp et al., 2013*; *Tian and Salis, 2015*). Genes encoded in the same mRNA are said to be translationally coupled when the translation of the upstream gene promotes translation of the downstream gene. Coupling was first reported in genetic studies in which nonsense mutations in an upstream gene suppress translation of a downstream gene (*Oppenheim and Yanofsky, 1980*). Such translational coupling is a conserved mechanism, observed in a number of genes across various bacterial species and their phages (*Berkhout and van Duin, 1985*; *van de Guchte et al., 1991*; *Govantes et al., 1998*; *Grabowska et al., 2011*). One of the key features regulating coupling efficiency is the distance between the stop codon of the upstream gene and the start codon of the downstream gene: distances less than 25 nt are optimal for coupling to occur (*Levin-Karp et al., 2013*; *Tian and Salis, 2015*). 75% of intergenic regions in polycistronic transcripts are shorter than this distance in *Escherichia coli* (*Yamamoto et al., 2016*), indicating that a high percentage of genes are optimally positioned for translational coupling.

How then does translational coupling occur? Two main mechanisms have been proposed in the literature: (1) ribosomes translating the upstream gene may unwind secondary structures that otherwise block de novo initiation at the downstream gene (*Baughman and Nomura, 1983*) or (2) following termination at the upstream stop codon and subunit splitting by RRF, 30S subunits may scan along the mRNA and re-initiate on the downstream gene without dissociating from the message (*Adhin and van Duin, 1990*; *Rex et al., 1994*). More recently, a model involving re-initiation by 70S post-TCs has also been proposed (*Yamamoto et al., 2016*). Given that loss of ribosome recycling should directly impact the level of mRNA-bound ribosomes capable of re-initiation (whether 30S or 70S), several studies inhibited RRF and monitored the efficiency of re-initiation on downstream genes, with mixed results. Part of the challenge is that RRF is encoded by an essential gene, complicating genetic analyses. Using a temperature-sensitive RRF allele, Kaji and co-workers reported higher levels of downstream gene expression on reporter constructs when RRF activity is reduced (*Janosi et al., 1998*), consistent with models of re-initiation by post-TCs. In contrast, using similar approaches, analysis of several coupled phage genes revealed no evidence that coupling depends on ribosome recycling (*Inokuchi et al., 2000*). Using a different strategy to deplete RRF (transcriptional shut-off with a ligand-inducible promoter), Nakamura and co-workers studied re-initiation after premature stop codons (i.e. nonsense mutations) within the *phoA* gene in *E. coli* and also argued against a role for RRF (*Karamyshev et al., 2004*). No clear picture emerges from these conflicting studies, and to our knowledge, the community has yet to characterize the role that RRF plays in the translational coupling of wild-type *E. coli* genes in their endogenous context within the genome. Context is critical because local mRNA structure plays such an important role in bacterial translational control.

Here, we report a genome-wide study of changes in translation that occur upon the loss of RRF in *E. coli*. We used ribosome profiling (deep sequencing of ribosome-protected mRNA fragments) (*Ingolia et al., 2009*) to determine the position of ribosomes throughout the transcriptome and monitor how ribosome density changes when recycling is inhibited. We established a method to rapidly and conditionally knock down RRF levels and collected samples at several time points after RRF depletion. We observe ribosomes accumulating upstream of stop codons, indicating that post-termination 70S complexes (post-TCs) fail to be recycled at stop codons and block elongating ribosomes

at the end of ORFs. We also see a dramatic accumulation of ribosome density in 3′-UTRs. We argue that these are not elongating ribosomes (translation in the 3′-UTR in reporter constructs is not enhanced by RRF depletion) but are post-TCs that have diffused away from the stop codon over time. Additionally, we observe that RRF depletion does not alter the ratio of ribosome density on neighboring genes in polycistronic transcripts, nor does it significantly alter the coupling efficiency in reporter assays of a series of *E. coli* genes previously demonstrated to be translationally coupled. These results argue that re-initiation by ribosomes or ribosome subunits bound to mRNA after recycling is not a widespread mechanism of translational initiation in *E. coli*. Finally, we observe that loss of recycling leads to significant changes in gene expression, including the accumulation of ribosome footprints on tmRNA and dramatic upregulation of ribosome rescue factor ArfA. Our results highlight the many critical roles played by RRF in translation in bacteria.

## Results

### Construction of the conditional RRF knock-down strain

Because RRF is encoded by an essential gene, previous analyses of the in vivo function of this factor have relied on a temperature-sensitive mutant (ts-RRF) whose abundance drops significantly at elevated temperatures (43°C) (*Janosi et al., 1998*). Given that heat shock introduces substantial changes in gene expression, we decided to take a different approach to conditionally deplete RRF to study its role in translation throughout the transcriptome (*Figure 1A*). To accomplish this, we swapped the promoter in the genome with the *araBAD* promoter so that cells cultured in media with arabinose express RRF, but following the switch to media with glucose, they strongly repress its transcription. In addition, we added a FLAG-epitope to the C-terminus of RRF to facilitate its detection, and the short peptide tag YALAA to target the protein for degradation by ClpXP (*Carr et al.,*

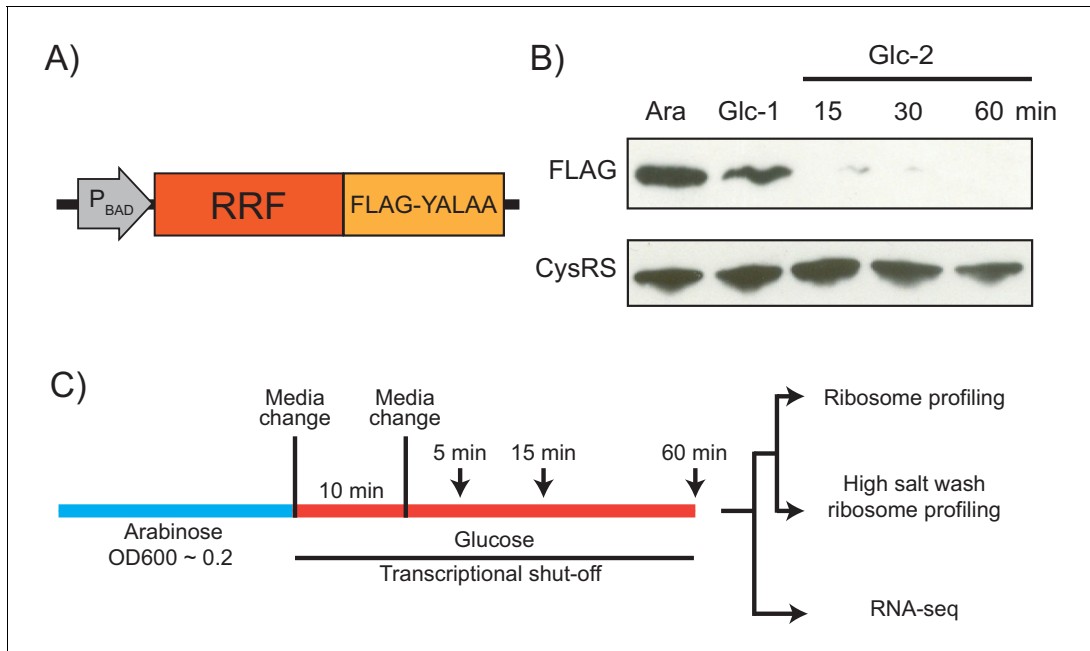

**Figure 1.** Strategy for ribosome recycling factor (RRF) depletion. (**A**) In the genome, the promoter upstream of *frr* (the gene encoding RRF) was replaced by the ligand-inducible *araBAD* promoter. A FLAG tag and residues YALAA were added to facilitate RRF detection and target it for degradation by ClpXP. (**B**) RRF depletion was monitored over time using antibodies against the FLAG epitope. CysRS serves as a loading control. The samples are: Ara (cultured in media with 0.2% arabinose), Glc-1 (harvested 10 min after the 1st change to media with 0.2% glucose), and three samples harvested 15, 30, and 60 min after the 2nd change to media with glucose. (**C**) Growth conditions for the samples used to make sequencing libraries. Note that the cultures were collected at earlier time points compared to (**B**).

The online version of this article includes the following figure supplement(s) for figure 1:

**Figure supplement 1.** Protein synthesis continues after ribosome recycling factor (RRF) knock down.

*2012*). In media containing arabinose, expression of RRF is sufficient to maintain viability even though the protein is rapidly turned over. In media containing glucose, the combination of transcriptional shut-off and accelerated protein degradation yield rapid depletion of the RRF protein: 10 min after the media switch, RRF levels are visibly reduced in immunoblots; 15 min following a second transfer to fresh media containing glucose, the protein is nearly undetectable (*Figure 1B*). To measure the levels of total protein synthesis after RRF depletion, we treated cells for 10 min with the methionine analog HPG beginning 5, 15, and 60 min after the second transfer to media containing glucose. We observe that bulk translation drops to about 40% by the first time point and then levels off (*Figure 1—figure supplement 1*). Even at the 60 min time point, there is still a substantial amount of protein synthesis. These data give us confidence that this approach can capture the effects of RRF depletion on ongoing protein synthesis and translational coupling.

Using this strategy, we performed ribosome profiling and RNA-seq on the wild-type (WT) and RRF knockdown (KD) strains over an extended time course (5, 15, and 60 min) after the second media switch (*Figure 1C*). In addition to the standard ribosome profiling libraries, we also prepared libraries from the same biological samples with a high-salt lysis buffer containing 1 M NaCl. High-salt conditions are known to dissociate ribosomes into 30S and 50S subunits unless the ribosomes are stabilized by the presence of an intact peptidyl-tRNA (*Zylber and Penman, 1970*). We anticipated that the loss of RRF would increase the number of post-termination 70S complexes (post-TCs). By preparing profiling libraries from the same samples with and without 1 M NaCl in the lysis buffer, we hoped to differentiate between elongating ribosomes and post-TCs following RRF depletion.

## Upon RRF depletion, ribosomes accumulate in the 3′-UTR and queue upstream of stop codons

Comparison of the ribosome profiling data from the WT and RRF KD strains reveals differences in ribosome density that are readily attributed to a diminution of ribosome recycling. For example, although in the WT strain ribosome protected footprints (RPFs) from the *rpsB* gene map almost exclusively to the coding sequence, after 60 min of RRF depletion, there is a clear accumulation of RPFs in the 3′-UTR (*Figure 2A*). These effects can be seen genome-wide in plots of ribosome density averaged over >1000 genes aligned at stop codons. (In these analyses, we excluded genes with ORFs within 110 nt downstream of the stop codon). Even after only 5 min of RRF depletion, ribosome density is higher in the 3′-UTR in the KD strain than it is in the WT strain (*Figure 2B*). The number of RPFs in the 3′-UTR is even higher after 15 min or 60 min of RRF depletion (*Figure 2C&D*). Because the ribosome profiling protocol involves the isolation of 70S monosomes on a sucrose gradient after digestion of unprotected mRNA by nucleases, we are confident that these represent 70S ribosomes and not 30S subunits. These data show that 70S ribosomes accumulate significantly in the 3′-UTR when recycling fails but cannot distinguish between actively translating ribosomes and post-termination complexes (post-TCs).

The loss of recycling also affects ribosome density within ORFs. We expected to see an increase in the height of the stop codon peak upon RRF depletion, due to the accumulation of post-TCs that are not recycled. We were surprised to find that the stop codon peaks are very similar in the WT and RRF KD strains (*Figure 2B–D*). We cannot explain why we do not see the expected increase in stop codon peaks upon RRF depletion, but we can infer that post-TCs accumulate there because we see stacked ribosomes appearing immediately upstream. The distance from the stop codon to the first upstream peak is about 25–30 nt, consistent with the footprint of a single ribosome, and a second peak is observed upstream at the distance of another footprint (*Figure 2B–D*). The density corresponding to these stacked ribosomes is more highly resolved in the ribosome profiling data collected using the high salt lysis buffer (*Figure 2F–H*). These data show that ribosomes stably bound at the stop codon prevent elongating ribosomes from reaching the end of the ORF as observed in *S. cerevisiae* upon depletion of Rli1 (*Young et al., 2015*). Taken together, our data show that RRF plays a critical role in clearing post-termination complexes in vivo, consistent with earlier in vitro biochemical studies.

## Ribosomes in the 3′-UTR are not translating

Prior studies using the ts-RRF allele in *E. coli* suggested that post-TCs can resume translation when recycling is disrupted (*Janosi et al., 1998*). We therefore asked if the ribosomes that accumulate in

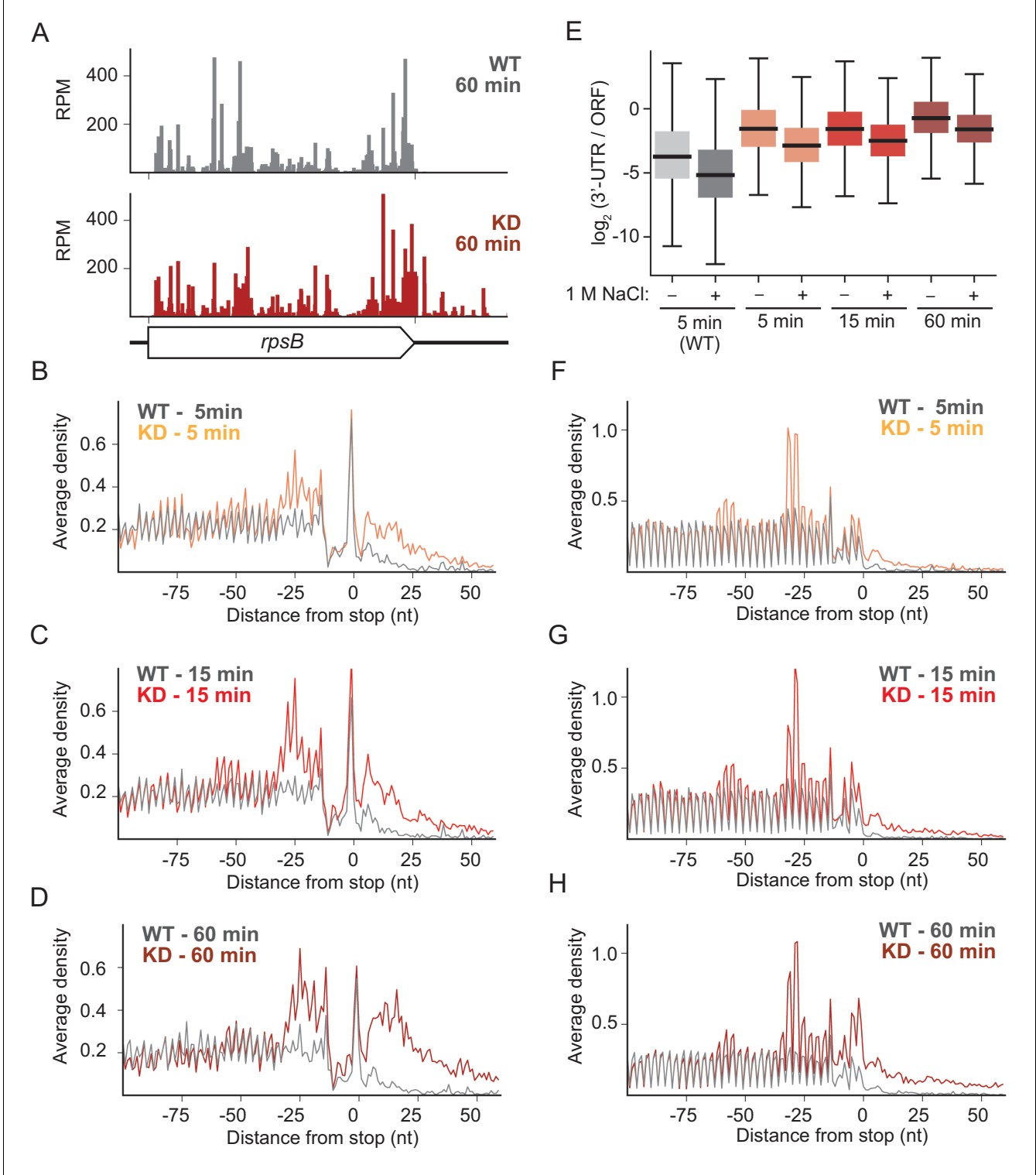

**Figure 2.** RRF knockdown creates queues of stalled ribosomes upstream of stop codons and high ribosome density in the 3'-UTR. (A) Ribosome density (in reads per million mapped reads) on the *rpsB* gene 60 min after the 2nd media change in the wild-type (top) and RRF knock-down strain (bottom). (B–D) Average ribosome density aligned at stop codons from standard ribosome profiling after 5 min (B), 15 min (C), and 60 min (D). (E) Ratio of ribosome density (in RPKM) in the 3'-UTR and upstream ORF, with and without high salt in the lysis buffer. (F–H) Average ribosome density at stop codons in libraries prepared with the same biological samples as (B–D) but with high-salt lysis buffer.

the 3′-UTR upon RRF knock-down are translating. High-salt concentrations are known to destabilize ribosomes lacking a nascent peptide (*Zylber and Penman, 1970*); we expect ribosome density corresponding to post-TCs to be reduced in the samples prepared with 1 M NaCl in the lysis buffer. Indeed, we see that the stop codon peak usually observed in standard RP experiments is lost in the high-salt samples, suggesting that this peak corresponds to post-TCs that are waiting for ribosome recycling. The fact that post-TCs accumulate to high levels at stop codons argues that ribosome recycling is slower than peptidyl-hydrolysis even in wild-type cells, consistent with earlier observations in *S. cerevisiae* (*Schuller et al., 2017*). In addition, we observe a reduction in the 3′-UTR ribosome density in the high-salt RP samples compared with the standard RP samples (compare *Figure 2B–D* with *Figure 2F–H*). The reduction of 3′-UTR ribosome density is quantified in *Figure 2E* comparing the ratio of ribosome density in 3′-UTR and the upstream ORF. For the 5 min time point, for example, the high-salt wash reduces 3′-UTR density by 2.5-fold (p-value from one-sided Mann-Whitney test, $1.1 \times 10^{-30}$). Again, the fact that 3′-UTR ribosomes are reduced by the high-salt lysis buffer argues that most of them are not translating but have moved into the 3′-UTR by diffusion. In contrast, the peaks corresponding to stacked ribosomes are not reduced by the high-salt conditions consistent with the idea that they represent elongating ribosomes whose intact peptidyl-tRNA stabilizes the subunits against dissociation in high salt.

Given that the high-salt buffer does not completely eliminate the 3′-UTR ribosome density, these profiling data cannot rule out the possibility that a low level of translation occurs in the 3′-UTR. To directly test this possibility, we designed reporters for two genes (*arcA* and *stpA*) that show high levels of ribosome density in the 3′-UTR when RRF is depleted. To detect re-initiation events, we inserted the GFP coding sequence into the 3′-UTR upstream of the first stop codon in every frame, creating three reporters, each in a different reading frame with respect to the upstream ORF (in-frame, +1 or −1) (*Figure 3A*). If post-TCs re-initiate in the 3′-UTR, GFP will be translated with a short peptide sequence on its N-terminus, depending on the initiation site and reading frame. Finally, we created a control construct in which the upstream ORF is replaced by the GFP gene (GFP-only), producing protein that can be used as a size marker in immunoblots and that reports on the level of transcription and translation from the native promoter and ribosome-binding sites.

In control experiments for the series of reporters based on *arcA*, we observe the GFP protein in the GFP-only construct using antibodies against GFP (*Figure 3B*) but not in a control strain expressing the empty vector without the GFP gene (EV in *Figure 3B*). Little to no GFP was observed in the RRF+ condition for reporters with GFP in any of the three frames, suggesting that there are normally very low levels of initiation at the site where we inserted GFP into the 3′-UTR. Upon RRF knockdown, even though GFP levels are higher from the GFP-alone construct relative to the CysRS loading control (perhaps indicative of an increase in RNA levels), no relative increase in GFP expression can be detected from the 3′-UTR in any reading frame. These data indicate that re-initiation does not occur at high levels upon RRF depletion. Likewise, for the series of reporters based on *stpA*, there was no increase in GFP expression in any of the reading frames upon RRF depletion. Together with the observation that much of the 3′-UTR ribosome density is lost upon high-salt treatment, these findings argue that re-initiation in the 3′-UTR is not a common event when RRF is depleted.

## Highly translated genes have more stacked ribosomes and relatively fewer 3′-UTR ribosomes

As shown above, when RRF is depleted, elongating ribosomes form a queue behind the stop codon (spaced one ribosome footprint apart) because they are sterically blocked by post-TCs that have failed to be recycled. We hypothesized that the number of stacked ribosomes would be particularly high on messages with high ribosome density. Using ribosome profiling and RNA-seq, we can estimate the ribosome occupancy of each transcript in the cell (the number of ribosome footprints per transcript, normalized by length). We selected two sets of genes corresponding to the top 20% and bottom 40% of ribosome occupancy (RO) values in the WT sample. For these two sets of genes, we calculated the average ribosome density aligned at stop codons after RRF depletion (*Figure 4A* and *Figure 4—figure supplement 1*). As anticipated, the high RO genes have significantly more stacked ribosomes, even at 5 min, with peaks corresponding to three or even four stacked ribosomes behind the post-TC complex at the stop codon. We quantified these effects by defining a collision score, the ratio of ribosome density in the last 100 nt of ORF to that of the entire ORF (using reads per kilobase per million mapped reads (rpkm) values that are normalized by length) (*Figure 4B*). We observe

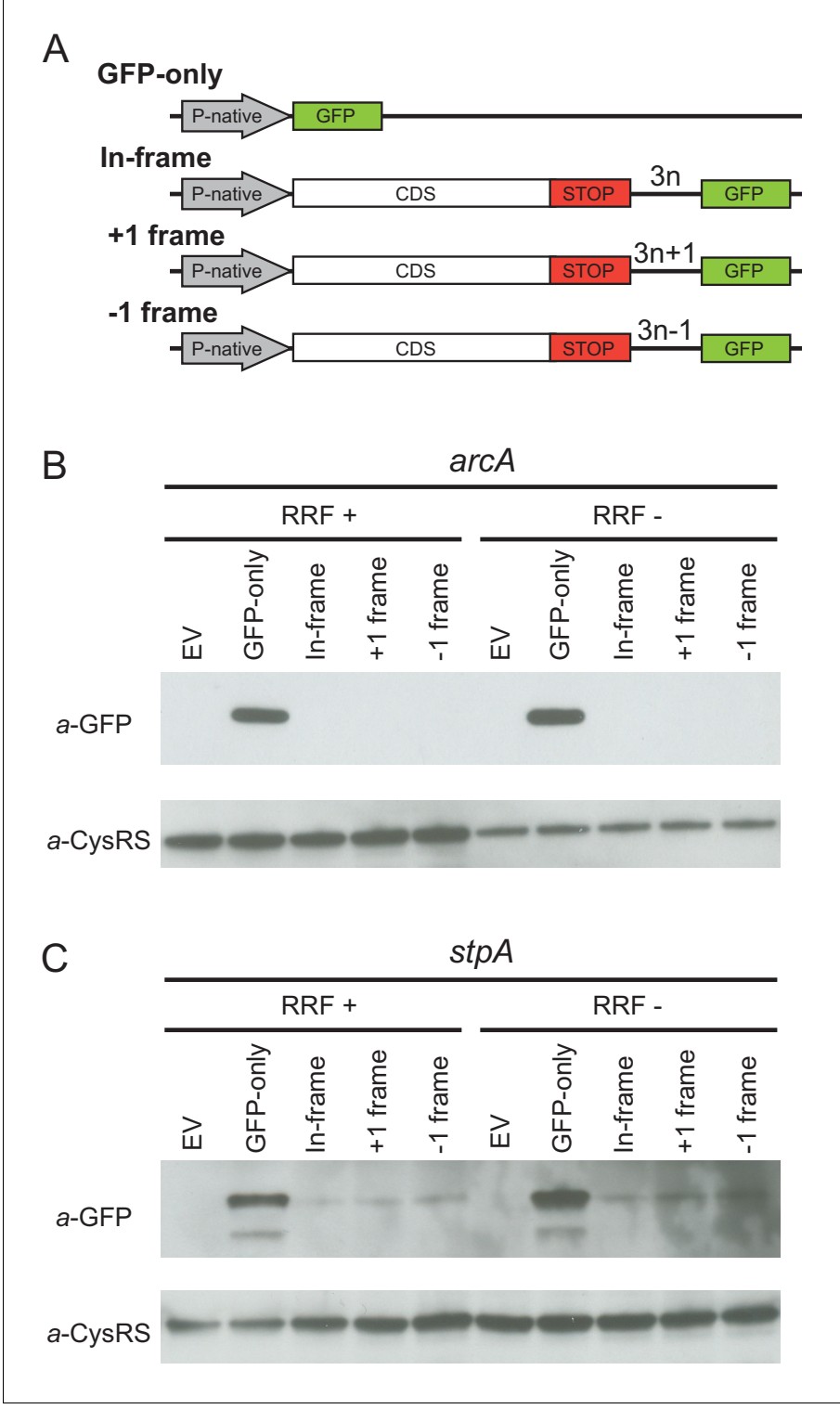

**Figure 3.** Ribosome recycling factor (RRF) depletion does not upregulate translation of GFP in the 3'-UTR of two genes. (**A**) The reporter plasmid encodes an *E. coli* gene with its native promoter, 5'-UTR and 3'-UTR. The GFP ORF is inserted into the 3'-UTR in each reading frame relative to the upstream ORF. The GFP-only serves as a control showing the level of expression from the native promoter and ribosome-binding site. Antibodies against GFP were used to observe GFP expression from the 3'-UTR of the *arcA* (**B**) and *stpA* genes (**C**) with and without 60 min of RRF depletion. EV corresponds to an empty vector control. CysRS serves as a loading control.

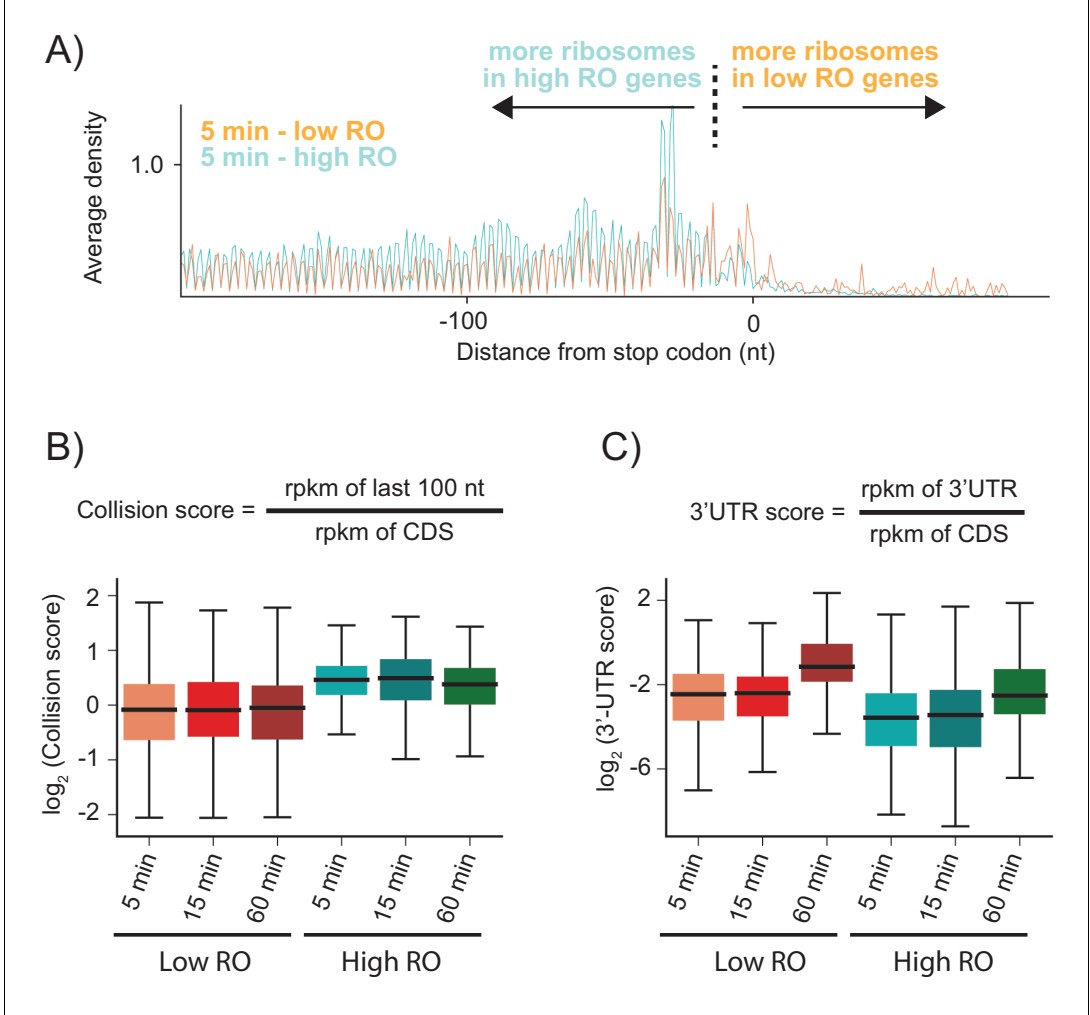

**Figure 4.** Highly translated genes have more stacked ribosomes and less relative ribosome density in the 3'-UTR. (**A**) High and low ribosome occupancy genes were selected using the ratio of ribosome profiling to RNAseq in the WT sample. Here, we show the average ribosome density aligned at stop codons using the high-salt data from the KD strain at the 5 min time point. (**B**) Evaluation of collision scores for genes with low and high ribosome occupancy (RO). (**C**) Evaluation of 3'-UTR scores for genes with low and high RO.

The online version of this article includes the following figure supplement(s) for figure 4:

**Figure supplement 1.** Average ribosome density aligned at stop codons of genes with low and high ribosome occupancy.

that for all three RRF KD samples (5, 15, and 60 min of depletion), the collision scores are higher for high RO genes than for low RO genes (*Figure 4B*, p-values from one-sided Mann-Whitney tests: $7.6 \times 10^{-23}$, $6.6 \times 10^{-16}$, $5.4 \times 10^{-15}$, respectively). These data show that because highly translated genes have more ribosomes per mRNA, there is an increased chance for collisions of elongating ribosomes with post-TCs trapped at stop codons.

Unexpectedly, genes with high RO have fewer ribosome footprints in the 3'-UTR relative to the CDS than genes with low RO (*Figure 4A*). To quantify this observation, we calculated the ratio of ribosome density in the 3'-UTR to the density in the upstream ORF (using rpkm values that are normalized by length), which we define as the 3'-UTR score (*Figure 4C*). At each time point, the 3'-UTR scores are lower in high RO genes than in low RO genes (*Figure 4C*). One possible explanation is that the stacking of ribosomes that occurs in high RO genes might stabilize post-TCs at the stop codon. In eukaryotes, ribosome collisions result in stable disome structures with extensive contacts between the 40S subunits (*Juszkiewicz et al., 2018*; *Ikeuchi et al., 2019*); similar interactions have also been observed in structures of bacterial polysomes (*Brandt et al., 2009*). Here, ribosome-

ribosome interactions or the binding of associated factors may prevent post-TCs in high RO genes from diffusing into the 3′-UTR.

## Loss of RRF does not affect translational coupling in polycistronic messages

One of the distinctive characteristics of bacterial genomes is that many genes are organized in polycistronic operons. For translationally coupled genes, translation of a downstream gene in an operon is dependent on the translation of an upstream gene (*Oppenheim and Yanofsky, 1980*). Following termination at the upstream stop codon and subunit splitting by RRF, 30S subunits may scan along the mRNA and re-initiate on the downstream gene without dissociating from the message (*Adhin and van Duin, 1990*). The global depletion of RRF and loss of ribosome recycling are expected to strongly impact translational coupling if re-initiation occurs with the same 30S subunit that translated the upstream gene. Without RRF and ribosome recycling, the number of scanning 30S subunits should be markedly reduced.

To determine what effect RRF may have in translational coupling on polycistronic mRNAs, we selected several gene pairs that have been directly demonstrated to exhibit translational coupling and asked how RRF depletion affects coupling efficiency in a reporter assay (*Figure 5A*). These gene pairs are: *trpB-trpA* (*Aksoy et al., 1984*), *galT-galK* (*Schümperli et al., 1982*), *prfA-prmC* (*Heurgué-Hamard et al., 2002*), *rpmI-rplT* (*Chiaruttini et al., 1996*), and *rplK-rplA* (*Baughman and Nomura, 1983*; *Figure 5B*). We constructed reporters in which the final 120 nt of the upstream ORF were cloned downstream of mCherry, followed by the natural intergenic region and the first 120 nt of the downstream ORF fused to nano-luciferase (nLuc). Two versions of each reporter were constructed: in one, the upstream gene contains a strong Shine-Dalgarno sequence (RBS), and in the other, the upstream gene lacks both a Shine-Dalgarno sequence and an AUG start codon (NoRBS). Because translation of the genes is thought to be coupled, the expression of the downstream gene (measured by nLuc activity) is expected to be higher for the RBS reporter than for the NoRBS reporter. As an internal control, we also express firefly luciferase (fLuc) from the reporter plasmid on a separate transcript. In the experiment, transcription of the nLuc reporters is induced at the second media change (*Figure 1C*) and their expression assayed after 60 min. The fact that we see robust expression of the nLuc reporters indicates that protein synthesis continues during this period of RRF depletion, making it possible to test the impact on translational coupling.

As expected, we observe translational coupling for all five reporter constructs: nLuc was expressed more strongly from the RBS reporter than from the NoRBS reporter in each case. The ratio of the nLuc/fLuc values for the RBS reporter and the NoRBS reporter reveals the degree of translational coupling. These ratios ranged for the five gene pairs from 1.7 for *trpB-trpA* to 7.0 for *rplK-rplA* (*Figure 5C*), validating that this assay measures translational coupling as designed. Importantly, the level of translational coupling did not change significantly upon RRF depletion (*Figure 5C*). These results with specific gene pairs in a reporter argue against models of translational coupling that involve re-initiation.

To look for effects of RRF depletion on the translation of neighboring genes genome-wide using ribosome profiling, we calculated the ratio of ribosome density on pairs of upstream and downstream genes, comparing the ratios in the WT and RRF KD strain. As shown for the *clpPX* operon in *Figure 6A*, even after 60 min of RRF depletion, where significant ribosome density accumulates in the untranslated region between the *clpP* and *clpX* genes, there are no discernable effects on the translation of the downstream *clpX* gene; the ratio of ribosome density (*clpP*/*clpX*) is very close to one both before and after RRF depletion. We then expanded this analysis to all pairs of upstream/downstream genes in polycistronic messages, as shown in *Figure 6B*, where each point represents a gene pair. There is a strong correlation (r = 0.89) between the ratio of ribosome density for the upstream and downstream genes between the RRF KD and the WT strains, indicating that the loss of RRF does not dramatically dysregulate the expression of the downstream genes across the transcriptome.

Despite this general lack of impact of RRF on translational coupling genome-wide, it is possible that this analysis misses strong effects on certain subsets of genes. We identified those sets of gene pairs that are most affected by RRF knockdown, showing either a twofold increase or decrease in their ratio of ribosome density (*Figure 6B*). We then asked if features known to affect translational coupling are enriched in these sets of genes. One such feature is the distance between the coupled

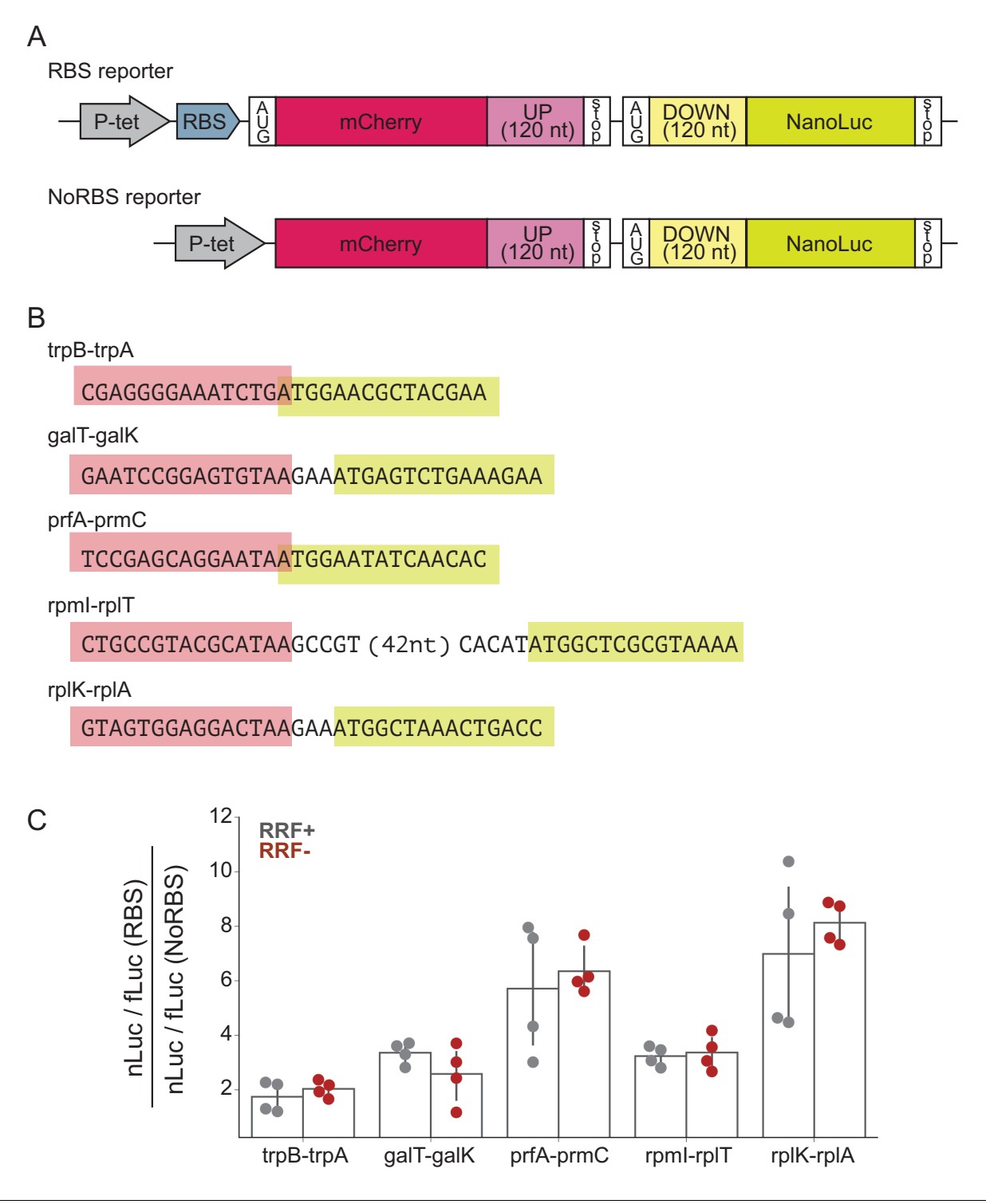

**Figure 5.** RRF depletion has little or no effect on five gene pairs known to be translationally coupled. (**A**) Reporter plasmids encode both mCherry and nano-luciferase separated by the last 120 nt of the upstream gene, the intergenic region, and the first 120 nucleotides of the downstream gene. The RBS reporter expresses mCherry from a strong Shine-Dalgarno motif whereas in the NoRBS reporter, mCherry lacks both an SD motif and a start codon. Firefly luciferase, an internal control, is expressed from the plasmid by an independent promoter. (**B**) Five gene pairs known to be translationally coupled were tested in the reporter assay. Expression was induced by the addition of anhydrotetracycline at the second media change (where RRF is already being depleted). (**C**) The ratio of nLuc expression from the RBS reporter/NoRBS reporter reports on the level of translational coupling with (grey) or without (red) RRF. The bar graph shows the mean and SD from four independent tests.

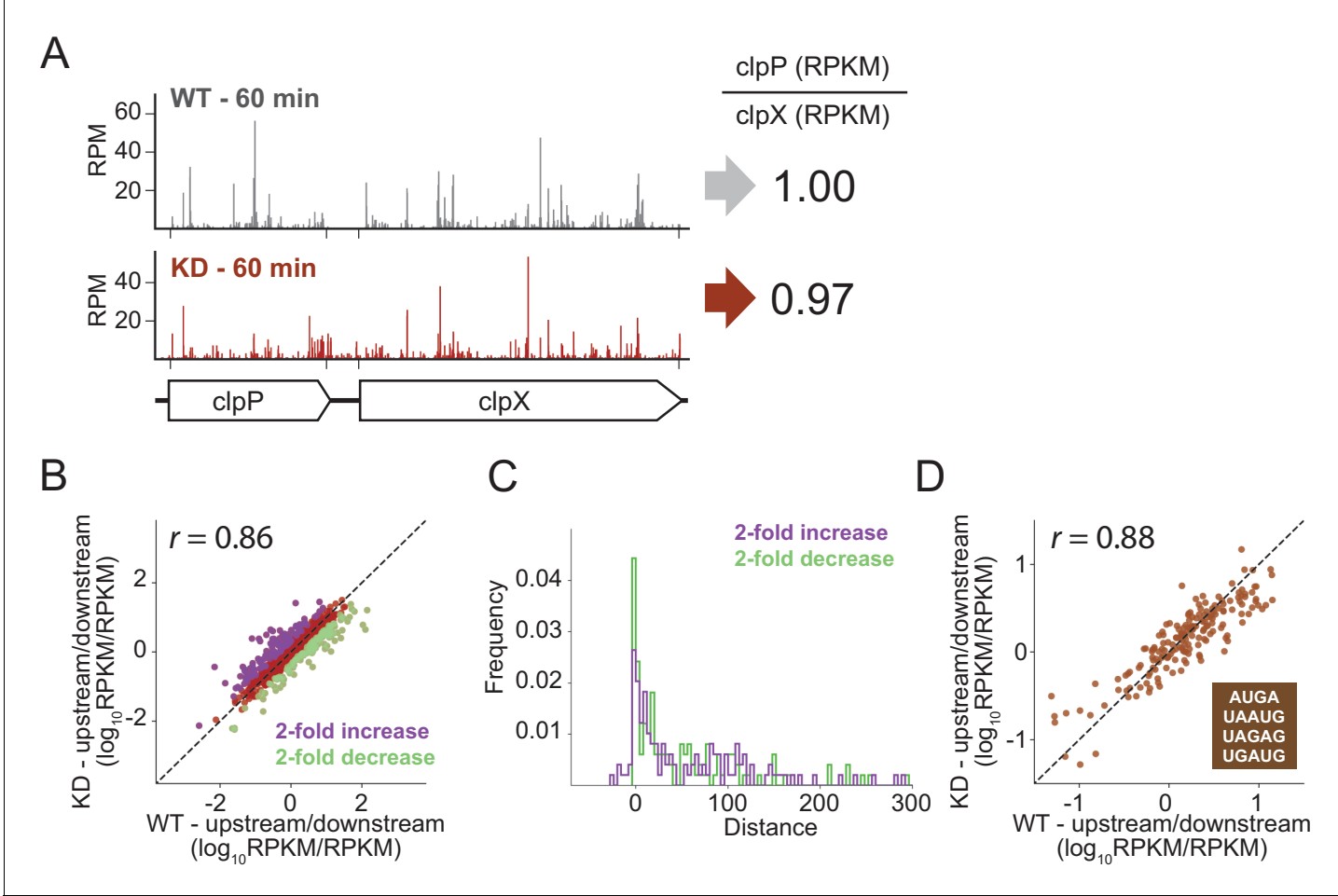

**Figure 6.** RRF depletion has little or no effect on the relative expression of gene pairs on polycistronic messages genome-wide. (A) Gene models showing the ribosome density (in reads per million mapped reads) on the *clpPX* operon 60 min after the second media change in the wild type (top) and RRF knock-down strain (bottom). The ratio of ribosome density (in RPKM) for the upstream/downstream gene is indicated to the right. (B) Scatter plot of the ratio of ribosome density for gene pairs (upstream/downstream) on polycistronic messages 60 min after the second media change in the wild-type (x-axis) and RRF knock-down strain (y-axis). Genes with a twofold increase or decrease in their ratio are colored purple and green, respectively. (C) Histogram of the intergenic distance for genes with twofold higher ratios upon RRF knock-down (purple) and genes with twofold lower ratios (green). (D) Scatter plot as in (B) but only including gene pairs with overlapping stop and start codons. *r* values indicate Pearson correlations in (B) and (D).

ORFs: generally, the longer the distance between the genes, the lower the efficiency of coupling (*Levin-Karp et al., 2013*; *Tian and Salis, 2015*). We compared the intergenic regions in the gene pairs whose ratios are twofold increased upon RRF depletion to those gene pairs that are twofold decreased; the two groups have distributions of intergenic lengths (*Figure 6C*) that are not significantly different statistically (p-value from one-sided Mann-Whitney test, 0.36).

We were also interested in pairs where the stop and start codons directly overlap, meaning that little or no scanning would be necessary, an optimal situation for re-initiation to take place. In *E. coli*, more than 30% of gene pairs have overlapping stop and start sites. We plotted the ratio of upstream and downstream ribosome density for these pairs and again observed that the WT and RRF-depleted strains are highly correlated (*Figure 6D*), arguing against the possibility that RRF plays a special role in these overlapping intergenic regions. Collectively, these data show that loss of ribosome recycling does not have a dramatic effect on translational coupling at the genome-wide level.

## Loss of ribosome recycling increases the demand for ribosome rescue

Finally, we asked what global changes in gene expression occur in response to the loss of ribosome recycling; these changes may offer insight into how cells respond or adapt to this stress. After 60 min of RRF depletion, we found that the number of ribosome footprints on 16 genes were more than 10-fold higher in the RRF KD strain than in the WT strain, indicating higher levels of protein synthesis (*Figure 7A*). One of the most prominent changes is in the expression of the ArfA protein (39-fold), a factor known to rescue stalled ribosomes (*Chadani et al., 2010*; *Chadani et al., 2012*). Although we observed a small increase in the levels of synthesis of ArfA protein after 15 min, the full 39-fold induction was observed after 60 min (*Figure 7B*). These changes in protein synthesis levels were accompanied by a corresponding up-regulation of the steady state levels of *arfA* mRNA (*Figure 7C*), arguing that the regulation acts at the RNA level. To test if this increase in mRNA level could be explained by higher levels of transcription, we created reporter plasmids with nano-luciferase (nLuc) driven by three promoters: the endogenous *arfA* promoter, a negative control with no promoter, or a positive control with the Tet promoter (*Figure 7D*). As expected, little or no nLuc expression was detected in the construct lacking a promoter, and the construct driven by the Tet promoter showed strong nLuc expression that was insensitive to the presence or absence of RRF (*Figure 7E*). Importantly, expression of nLuc from the *arfA* promoter did not change significantly upon RRF depletion, arguing against a strong induction of transcription and leading us to look for other explanations for the dramatic increase in the steady-state level of *arfA* mRNA.

The expression of *arfA* is regulated post-transcriptionally based on the demand for ribosome rescue activity in the cell (*Garza-Sánchez et al., 2011*). *arfA* mRNA contains a hairpin structure that is constitutively cleaved by RNase III leading to the production of a processed mRNA lacking a stop codon (a 'non-stop' mRNA). Ribosome stalling at the 3'-end of this truncated *arfA* transcript is resolved by the primary ribosome rescue factor, tmRNA, which tags the truncated ArfA protein to target it for degradation. tmRNA also recruits RNase R so that the *arfA* mRNA is rapidly degraded (*Richards et al., 2006*). If tmRNA is overwhelmed or inactive, ArfA expression increases and together with RF2 provides a backup mechanism to rescue stalled ribosomes (*Chadani et al., 2010*; *Chadani et al., 2012*). Based on this, we suspected that the increase in *arfA* mRNA levels that we observe is explained by inhibition of tmRNA activity and reduced rates of mRNA decay by RNase R.

Given this model, why is tmRNA activity inhibited when RRF is depleted? After only 5 min of RRF depletion, we see a sharp increase in ribosome footprints on the short ORF within tmRNA (*Figure 7F*). The strongest peak is at the stop codon, suggesting an accumulation of post-TCs that cannot be recycled. 60 min after depletion of RRF, we also see an accumulation of ribosome footprints downstream of the tmRNA ORF, in regions that are normally not translated but play important structural roles. These downstream footprints are reminiscent of the accumulated 3'-UTR ribosomes in all mRNAs and likely represent non-translating 70S ribosomes; consistent with this explanation, there is less ribosome density in tmRNA in the high-salt treated samples where post-TCs are destabilized (data not shown). We speculate that these post-TCs denature key structures in tmRNA, interfering with its ability to rescue stalled ribosomes and thus preventing it from promoting degradation of ArfA at the mRNA and protein levels. These findings reveal how a translation-based feedback pathway which responds to aberrant cellular translation is induced by the loss of ribosome recycling.

## Discussion

In this report, we provide a global view of the effects of loss of ribosome recycling on protein synthesis in *E. coli*. In contrast to previous studies of RRF function in vivo that utilized a temperature sensitive allele of RRF, we developed a strategy relying on transcriptional shutoff and rapid protein turnover through targeted proteolysis (*Figure 1A*). This strategy avoids shifts in temperature that induce global stress responses. There are a couple of caveats, however. First, our approach probably does not lead to total loss of this essential factor; low levels of recycling likely occur in RRF-depleted cells. A second caveat is that RRF depletion may lead to indirect effects. Biological pathways unrelated to recycling may be impacted when the translation of critical proteins is reduced. To minimize the impact of potential indirect effects, we focus our discussion on aspects of ribosome activity that are directly tied to recycling. In addition, most of the phenotypes of RRF depletion described here occur even at the earliest time point (5 min), where indirect effects are most unlikely.

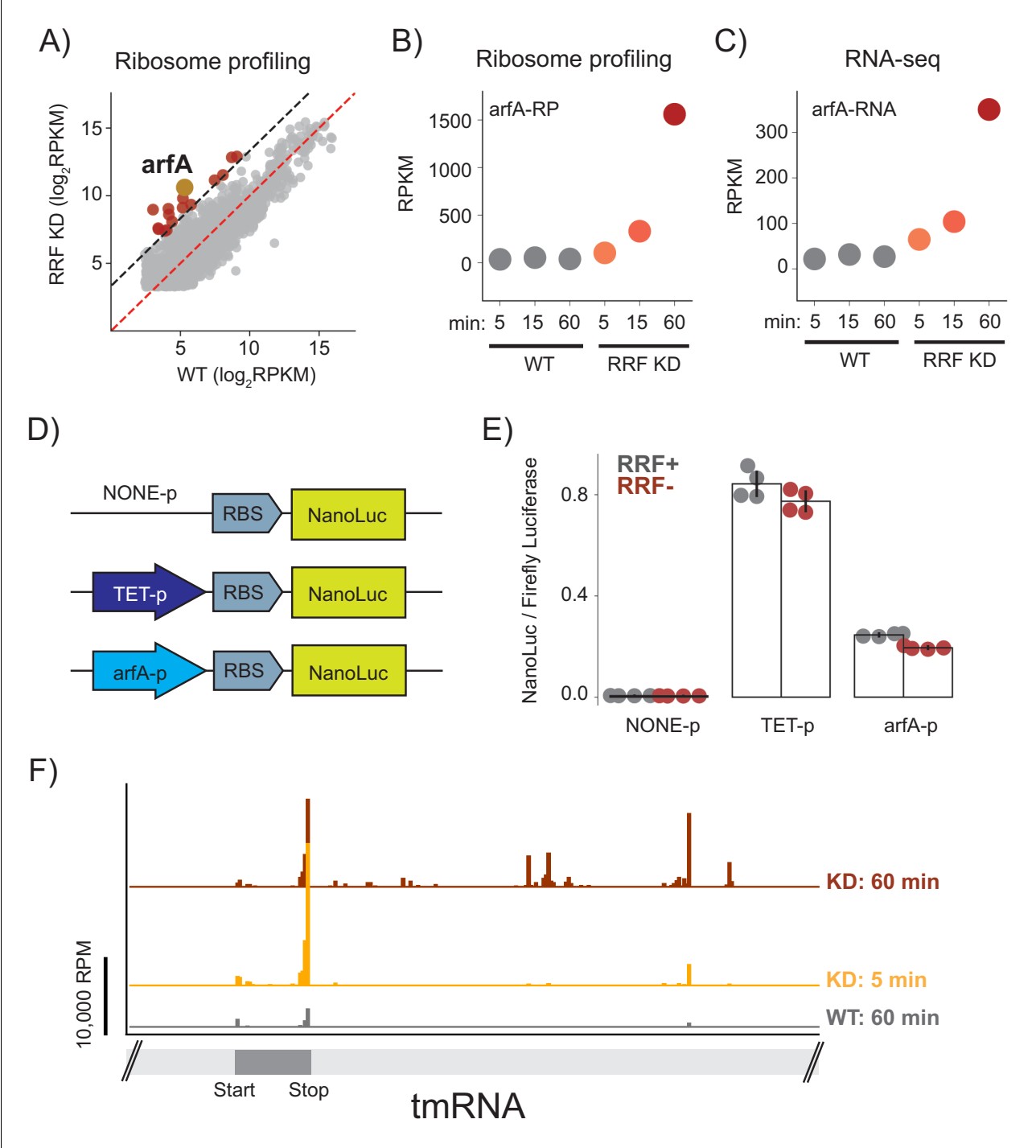

**Figure 7.** Ribosome recyclinf factor (RRF) depletion upregulates ArfA expression. (A) Scatter plot of ribosome density (in rpkm) in the wild type (x-axis) and RRF knock-down strain (y-axis) 60 min after the second media change. Genes with 10-fold increase of ribosome density are colored red; *arfA* is highlighted in yellow. (B) Levels of *arfA* translation from ribosome profiling at each time point. (C) Levels of *arfA* mRNA from RNA-seq at each time point. (D) Reporter assay to measure transcription from the *arfA* promoter. Nano-luciferase is expressed from a plasmid using the same RBS either without a promoter, from the TET-promoter, or from the *arfA* promoter. Firefly luciferase, an internal control, is expressed from the plasmid by an independent promoter. (E) The nLuc/fLuc ratio reports on the level of transcription by the three promoters with (grey) or without (red) RRF of promoter assay. The bar graph shows the mean and SD from four independent tests. (F) Ribosome density (in reads per million mapped reads) on part of the gene for tmRNA, including the short region that is translated.

Given the well-characterized role of RRF in splitting ribosomes following termination, we expected to see a strong accumulation of post-termination complexes (post-TCs) at stop codons across the transcriptome upon depletion of RRF. Although we did not see differences in stop codon peaks, we did observe indirect evidence that post-TCs are in fact present at stop codons. Elongating ribosomes form a queue behind stop codons, spaced one ribosome footprint apart. We infer that post-TCs accumulate at stop codons, blocking elongation and trapping upstream ribosomes in an inactive state that prevents them from completing protein synthesis. We speculate that post-TCs retain some affinity for the site where termination occurred due to base-pairing between the deacyl-tRNA in the P-site and the last sense codon in the ORF. These findings are consistent with prior observations of stacked ribosomes that appear upon knockdown of the yeast 80S recycling factor Rli1 (*Young et al., 2015*) and the 40S recycling factors Tma64, Tma22, and Tma20 (*Young et al., 2018*). These data show that RRF is critical for ribosome recycling at stop codons and that the loss of recycling interferes with robust protein synthesis from an mRNA.

Our second main observation is the accumulation of non-translating 3′-UTR ribosomes upon RRF depletion. Unlike the stacked ribosomes upstream of stop codons, which contain peptidyl-tRNA, ribosomes in the 3′-UTR are lost in high-salt concentrations, arguing that they are post-TCs and not elongating 70S ribosomes. How did they get into the 3′-UTR? We argue that over time, post-TCs, only weakly retained by codon-anticodon pairing with the tRNA, eventually slide along the mRNA past the stop codon into the 3′ UTR. The dramatic enrichment of ribosome density downstream of stop codons highlights the importance of RRF in maintaining ribosome homeostasis by regenerating ribosome subunits to initiate translation. When RRF is depleted, many ribosomes are trapped in an inactive state in the 3′-UTR, as seen in earlier studies (*Guydosh and Green, 2014*; *Mills et al., 2016*).

How are post-TCs in the 3′-UTR removed in the absence of the canonical, RRF-mediated recycling pathway? From prior work we have some idea of what happens to empty 80S ribosomes in eukaryotes. The Dom34/Hbs1 complex (normally involved in rescuing stalled ribosomes) delivers Dom34 to the ribosomal A site, where it works together with the canonical recycling factor Rli1 to promote subunit splitting (*Shoemaker and Green, 2011*). In *S. cerevisiae*, loss of Dom34 leads to the accumulation of non-translating ribosomes in the 3′-UTR, indicating that Dom34 normally helps remove empty 80S ribosomes from 3′-UTRs (*Guydosh and Green, 2014*). Because the Dom34/Rli1 complex directly splits ribosomal subunits whether there is an intact peptidyl-tRNA linkage or not, it can serve as a backup mechanism to recycle post-TCs that have escaped canonical recycling at the stop codon by eRF1/Rli1 (*Young et al., 2015*). In contrast, the bacterial tmRNA/SmpB and ArfA/RF2 ribosome rescue systems typically catalyze reactions with the peptidyl-tRNA. It may be that bacteria lack an effective backup mechanism for splitting post-TCs. The critical role that RRF plays in ribosome homeostasis likely is the simplest explanation for its essential nature.

Finally, we return to a fundamental and important difference in the role of recycling revealed by our studies. In yeast, ribosome density in the 3′-UTR upon knockdown of Rli1 or 40S recycling factors at least in part reflects the activity of elongating ribosomes (*Young et al., 2015*; *Young et al., 2018*). Although the mechanism is poorly understood, translation in the 3′-UTR does arise from re-initiation downstream of the stop codon. In contrast, the 3′-UTR ribosome density in *E. coli* reflects post-TCs that are not translating, as evidenced by their sensitivity to the high-salt buffer (*Figure 2*) and our failure to detect an increase in translation of GFP in the 3′-UTR when RRF is depleted (*Figure 3*). Although the reason for these differences between *S. cerevisiae* and *E. coli* are not clear, these findings appear to be consistent with the different initiation strategies used by eukaryotes and bacteria. In general, eukaryotic ribosomes scan from the 5′-end of mRNAs to find start codons (*Hinnebusch, 2014*). Given the importance of short, upstream ORFs in regulating translation in eukaryotes, ribosomes must be able to re-initiate following termination (*Gunišová et al., 2018*). On the other hand, bacterial 30S subunits normally initiate anywhere along a polycistronic mRNA where they are recruited directly based on sequence context and mRNA structure (*Salis et al., 2009*; *Del Campo et al., 2015*; *Burkhardt et al., 2017*; *Baez et al., 2019*; *Saito et al., 2020*), without the need for scanning.

The fact that we do not observe robust re-initiation upon loss of ribosome recycling in *E. coli* has important implications for models of translational coupling. One common mechanistic model argues that following termination and subunit splitting, the 30S subunit remains on the mRNA and slides until it re-initiates at a nearby start codon (*Adhin and van Duin, 1990*). If this model were correct,

we reasoned that recycling defects should reduce the number of such scanning 30S subunits downstream of stop codons, thereby *decreasing* the expression of downstream genes. An alternate re-initiation mechanism involving 70S ribosomes was proposed by Nierhaus and co-workers (*Yamamoto et al., 2016*). We note that initiation by 70S ribosomes is difficult to reconcile with the well-characterized biochemical activities of fMet-tRNA and initiation factors that initiate translation on 30S subunits (*Simonetti et al., 2009*). Nevertheless, this model also predicts that recycling defects should *increase* the efficiency of re-initiation upon RRF depletion (*Figure 2*). Contrary to the predictions of both of these models, however, we observed that the loss of RRF does not affect the relative translational levels of neighboring genes in polycistronic messages. This was true in reporters based on five gene pairs previously shown to be translationally coupled (*Figure 5*) as well as in genome-wide analyses (*Figure 6*). We conclude that re-initiation is not the dominant mechanism for translational coupling in *E. coli*, although we cannot rule out the possibility that it may occur in specific contexts.

Instead, our data are most consistent with models of de novo initiation, where free 30S subunits are recruited to start codons directly. In cases where neighboring genes are translationally coupled, it seems likely that melting of mRNA structure near the downstream start site is the mechanism at play. The melting of mRNA structures is a well-documented mechanism for coupling the translation of neighboring genes where it has been explored in detail (*Rex et al., 1994*; *Chiaruttini et al., 1996*). Moreover, several genome-wide studies of mRNA structure and translation argued that mRNA structure is a major determinant of translational efficiency in bacteria (*Del Campo et al., 2015*; *Burkhardt et al., 2017*; *Mustoe et al., 2018*), perhaps even more important than well-studied sequence elements such as Shine-Dalgarno motifs (*Saito et al., 2020*). The ability to determine secondary structures of mRNA in vivo in a high-throughput fashion will be a powerful method to inform future studies of the mechanism of translational coupling in bacteria.

## Materials and methods

### Growth conditions

For ribosome profiling and RNA-seq libraries, cells were cultured at 37°C in 500 mL of LB + arabinose (0.2% w/v) to $OD_{600}$ = 0.2. Then, cells were collected by centrifugation, resuspended in 500 mL of LB + glucose (0.2% w/v), and cultured at 37°C for 10 min. Cells were collected by centrifugation, resuspended in 500 mL of LB + glucose (0.2% w/v) a second time and cultured at 37°C for 5, 15, or 60 min.

### Measurement of bulk translation levels

Levels of protein synthesis were measured by HPG incorporation (*Sherratt et al., 2017*; *An et al., 2020*). Cultures were grown at 37°C in M9 media containing 1x M9 salt (49 mM $Na_2HPO_4$, 22 mM $KH_2PO_4$, 8.6 mM NaCl, 18.7 mM $NH_4Cl$), 100 µM $CaCl_2$, 1 mM $MgSO_4$, 1x minimal essential medium (MEM) vitamin solution, 1x methionine biosynthesis inhibition amino acids (L-lysine (100 µg/ml), L-threonine (100 µg/ml), L-phenylalanine (100 µg/ml), L-isoleucine (50 µg/ml), L-leucine (50 µg/ml), and L-valine (50 µg/ml)), and 0.2% carbon source. To monitor bulk translation rate in arabinose media, HPG (50 µg/ml) was added to a culture in arabinose media at $OD_{600}$ = 0.5, and cells were incubated for 10 min. To determine the background level of the Alexa Fluor 488 signal, methionine (50 µg/ml) was added (instead of HPG) in arabinose media and the cells were incubated for 10 min. For RRF depletion, cells were transferred to glucose media at $OD_{600}$ = 0.5, and after another 10 min, a second change to fresh glucose media was performed. At the indicated time points after the second media change (5, 15, 60 min), HPG (50 µg/ml) was added and the cells were further incubated for 10 min. Cells were lysed with Click-iT lysis buffer (1% SDS in 50 mM Tris-HCl, pH 8.0). Total protein concentration was measured with the DC protein assay (Bio-Rad). 15 µg of lysate in 30 µl Click-iT lysis buffer was first mixed with 15 µl of 4 × Click master mix (4 mM $CuSO_4$, 4 mM sodium ascorbate, 400 µM TBTA ligand in Click-iT lysis buffer), then with 15 µl of 400 uM Alexa Fluor 488 Azide (Thermo Fisher) in Click-iT lysis buffer. The Click-iT reaction mixture was incubated in the dark for 2 hr. Unreacted Alexa Fluor 488 Azide was removed by $CHCl_3$/MeOH precipitation. Samples were resolved by SDS–PAGE and the Alexa Fluor 488 signal was detected by Typhoon FLA9500 (GE

Healthcare). Total protein was visualized by Coomassie brilliant blue. The signals were quantified by ImageJ (*Schneider et al., 2012*).

## Cell harvesting and lysis

Cells were harvested by filtration using a Kontes 99 mm filtration apparatus and 0.45 µm nitrocellulose filter (Whatman) and then flash frozen in liquid nitrogen. Cells were lysed in lysis buffer (20 mM Tris pH 8.0, 10 mM $MgCl_2$, 100 mM $NH_4Cl$, 5 mM $CaCl_2$, 100 U/mL DNase I, and 1 mM chloramphenicol) or 1 M NaCl lysis buffer (20 mM Tris pH 8.0, 10 mM $MgCl_2$, 100 mM $NH_4Cl$, 1 M NaCl, 5 mM $CaCl_2$, 100 U/ml DNase I, and 1 mM chloramphenicol) using a Spex 6870 freezer mill with 5 cycles of 1 min grinding at 5 Hz and 1 min cooling. Lysates were centrifuged at 20,000 x g for 30 min at 4°C to pellet cell debris. To exchange the buffer prior to nuclease digestion, samples in the 1 M NaCl lysis buffer were layered on a 1 mL sucrose cushion (20 mM Tris pH 7.5, 500 mM $NH_4Cl$, 0.5 mM EDTA, 1.1 M sucrose), centrifuged by a TLA 100.3 rotor at 65,000 rpm for 2 hr, and resuspended in the standard lysis buffer.

## Library preparation

10–54% sucrose density gradients were prepared using the Gradient Master 108 (Biocomp) in the gradient buffer (20 mM Tris pH 8.0, 10 mM $MgCl_2$, 100 mM $NH_4Cl$, 2 mM DTT). 5–20 AU of *E. coli* lysate was loaded on top of sucrose gradient and centrifuged in a SW41 rotor at 35,000 rpm for 2.5 hr at 4°C. Fractionation was performed on a Piston Gradient Fractionator (Biocomp). Libraries for ribosome profiling and RNA-seq are prepared as in *Woolstenhulme et al., 2015*; *Mohammad et al., 2016*, analyzed on a BioAnalyzer high sensitivity DNA kit (Agilent), and sequenced on the HiSeq2500 (Illumina).

## Western blots

Cells were grown as described above and harvested by centrifugation 60 min after the second resuspension in LB + glucose (0.2% w/v). After centrifugation, cells were resuspended in 12.5 mM Tris pH 6.8, 4% SDS and heated to 90°C for 10 min. 5x loading dye (250 mM Tris pH 6.8, 20% glycerol, 30% β-mercaptoethanol, 10% SDS, saturated bromophenol blue) was added and lysate was heated to 90°C for 10 min. Protein was separated by 4–12% Criterion XT Bis-Tris protein gel (Bio-Rad) and XT MES buffer and transferred to PVDF membrane by Trans-Blot Turbo Transfer system (Bio-Rad). Membranes were blocked in 5% milk for 1 hr at room temperature. The membranes were probed by antibodies diluted in TBS-tween. FLAG-tagged proteins were detected by anti-FLAG-HRP in 1:10,000 dilution (Sigma). GFP was detected by anti-GFP in 1:2000 (Takara) and anti-mouse-HRP in 1:10,000 (Thermo Fisher). Cysteinyl-tRNA synthetase (CysRS) was detected by anti-CysRS in 1:2000 (from Dr. Ya-Ming Hou) and anti-rabbit-HRP in 1:4000 (from Dr. Ya-Ming Hou). Chemiluminescent signals of HRP were developed by SuperSignal West Pico PLUS Chemiluminescent Substrate (Thermo Fisher) or SuperSignal West Femto Maximum Sensitivity Substrate (Thermo Fisher), and exposed on Amersham Hyperfilm ECL (GE).

## Luciferase assays

Cells were grown as described for the ribosome profiling and RNA-seq libraries. At the second media change to fresh LB + glucose, anhydrotetracycline (1 nM) was added and the cells were harvested after another 60 min. Of cultured samples, 45 µl were mixed with 5 µl of phosphate buffer (1 M $K_2HPO_4$ pH 7.8 and 20 mM EDTA) and frozen on dry-ice. Frozen samples were mixed with 150 µl of Luciferase assay lysis buffer (Cell Culture Lysis Reagent (Promega), 1.25 mg/ml lysozyme, and 2.5 mg/ml BSA) and incubated for 10 min at room temperature. Nanoluc and firefly luciferase activity were detected by Nano-Glo Dual-Luciferase Reporter Assay System (Promega) according to the manufacturer's protocol. Chemiluminescence was monitored by G:BOX (Syngene).

## General processing of sequencing data

The adaptor sequence CTGTAGGCACCATCAAT was removed by Skewer (*Jiang et al., 2014*). Reads mapping to tRNA and rRNA were removed; the remaining reads to aligned to *E. coli* MG1655 genome build NC_000913.2 using bowtie version 1.1.2 (*Langmead et al., 2009*). The position of the ribosomes was assigned using the 3′-end of aligned reads. 3′-UTRs were defined using

the transcript units determined by Wanner and co-workers (*Conway et al., 2014*). Polycistrons for analyses of translational coupling are based on the same transcript units. In cases where a gene is assigned to multiple transcription units, we selected the one annotated in RegulonDB as an operon. In addition, we excluded pairs of upstream and downstream genes where their RNA abundances (calculated from RNA-seq) differ by more than 5-fold. 3′-UTR scores were calculated as the RPKM of 3′-UTR divided by RPKM of the upstream CDS. Collision scores were calculated as the RPKM of the last 100 nt of CDS divided by RPKM of the entire CDS. In RPKM calculattions, reads mapped on the last 15 nucleotides of the CDS are excluded. RO is the RPKM of ribosome profiling reads for each gene divided by RPKM of RNA-seq.

## Acknowledgements

The authors thank David Mohr at the Genetics Resources Core Facility, Johns Hopkins Institute of Genetic Medicine, for sequencing assistance. This study was funded by a JSPS fellowship (KS), NIH grant GM110113 (ARB), and HHMI (RG).

## Additional information

### Competing interests

Rachel Green: Reviewing editor, *eLife*. The other authors declare that no competing interests exist.

### Funding

| Funder | Grant reference number | Author |
| --- | --- | --- |
| National Institute of General Medical Sciences | GM110113 | Allen R Buskirk |
| Howard Hughes Medical Institute | | Rachel Green |
| Japan Society for the Promotion of Science | JSPS fellowship | Kazuki Saito |

The funders had no role in study design, data collection and interpretation, or the decision to submit the work for publication.

### Author contributions

Kazuki Saito, Conceptualization, Formal analysis, Investigation, Methodology, Writing - original draft; Rachel Green, Conceptualization, Funding acquisition, Writing - review and editing; Allen R Buskirk, Conceptualization, Formal analysis, Supervision, Funding acquisition, Writing - review and editing

### Author ORCIDs

Rachel Green (iD) http://orcid.org/0000-0001-9337-2003
Allen R Buskirk (iD) https://orcid.org/0000-0003-2720-6896

### Decision letter and Author response

Decision letter https://doi.org/10.7554/eLife.59974.sa1
Author response https://doi.org/10.7554/eLife.59974.sa2

## Additional files

### Supplementary files

• Transparent reporting form

### Data availability

Sequencing data have been deposited in GEO under accession codes GSE151688.

The following dataset was generated:

| Author(s) | Year | Dataset title | Dataset URL | Database and Identifier |
|---|---|---|---|---|
| Buskirk AR | 2020 | RRF plays critical roles in ribosome homeostasis in *E. coli* but has little effect on translational coupling | https://www.ncbi.nlm.nih.gov/geo/query/acc.cgi?acc=GSE151688 | NCBI Gene Expression Omnibus, GSE151688 |

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
