## [Decision Letter]

**Acceptance summary:**

Following translation termination in bacteria, ribosomes must be split into the large and small subunits by a recycling factor. Saito et al. show that depletion of recycling factor in *Escherichia coli* leads to traffic jams of translationally active ribosomes upstream of stop codons, and diffusion of translationally inactive ribosomes downstream of stop codons. Moreover, Saito et al. provide strong evidence that terminating ribosomes cannot reinitiate translation on a downstream gene on the same RNA, arguing against two major models for translational coupling.

**Decision letter after peer review:**

Thank you for submitting your article "RRF is critical for ribosome recycling and homeostasis in *E. coli* but not for translational coupling genome-wide" for consideration by *eLife*. Your article has been reviewed by three peer reviewers, including Joseph T Wade as the Reviewing Editor and Reviewer #3, and the evaluation has been overseen by Gisela Storz as the Senior Editor. The following individuals involved in review of your submission have agreed to reveal their identity: Pavel V Baranov/Patrick O'Connor (Reviewer #2).

The reviewers have discussed the reviews with one another, and the Reviewing Editor has drafted this decision to help you prepare a revised submission.

Summary:

The reviewers found the work to be of high quality, and agreed that it represents an important advance in our understanding of both RRF function and translational coupling. Our one significant concern is that the bulk translation rate after RRF depletion was not measured. Interpretation of data from several experiments looking at translation coupling relies on the assumption that translation is ongoing in the RRF-depleted cells. The reviewers have a number of other suggestions that can all be addressed without the need for further experiments.

Essential revisions:

1) Measure the bulk translation rate across the time-course of RRF depletion. The current interpretation of Figure 3, Figure 5 and Figure 6 is based on the assumption that translation is ongoing in the RRF-depleted cells. If translation levels are greatly reduced, the conclusions about translational coupling might not be valid.

2) Acknowledge the fact that residual recycling is likely occurring in the RRF-depleted cells. If it is possible to measure this residual activity, that would be great, but such an experiment is not required.

3) Include some discussion of the potential impact of pleiotropic effects of RRF depletion on the observed phenotypes. Given that most of the phenotypes were observed at the 5 minute time-point, the reviewers did not see this as a critical problem.

4) Change the title to focus solely on translational coupling, which in the opinion of the reviewers is the most important aspect of the paper.

The reviewers have additional comments, detailed below, that you should respond to with changes to the text where appropriate.

Reviewer #1:

Summary:

Although this MS addresses the interesting question of RRF function in the translation of coupled transcripts in *E. coli*, it depends upon the correctness of two fundamental assumptions that are not examined critically in the text. The first assumption is that effects resulting from the absence of RRF protein, which is almost complete at 15' and complete at 60' (subsection “Upon RRF depletion, ribosomes accumulate in the 3’-UTR and queue upstream of stop codons”), can be solely attributed to the loss of RRF protein. Given that rrf is an essential gene, RRF removal could well have pleiotropic effects not considered by the authors. The second, and related, assumption is that ribosome occupancy is a faithful reporter of protein translation rates when RRF is absent. Missing are any direct measures of protein synthesis and cell viability that would test these assumptions.

Essential revisions:

1) The present Discussion section is overlong and very speculative. It would be improved by substantial condensation. On the other hand, a critical analysis of the shortcomings of prior results on translational coupling and RRF would be valuable.

2) Subsection “Loss of RRF does not affect translational coupling in polycistronic messages”. First paragraph. The last sentence is unclear. Also, it would be helpful to the reader to include a description of the essence of the Yamamoto et al., proposed mechanism. This might be better placed in the Discussion section.

3) Figure 7F should be redrawn. At present, it is very reader unfriendly

Reviewer #2:

In this study, Saito et al., explored the role of RRF by examining how mRNA translation responses to RRF knockdown in *E. coli* using ribosome profiling and a series of ad hoc expression reporters. The major (somewhat surprising) findings are that RRF KD has relatively little effect on translational coupling and that RRF KD triggers upregulation of ArfA ribosome rescue pathway via inactivation of tmRNA by unrecycled ribosomes. This work is of an outstanding technical quality and is very important for our understanding of ribosome recycling in bacteria and of the mechanism of translational coupling.

Essential revisions:

1) In the absence of direct evidence, the authors are reasonably cautious not to state that translation re-initiation does not take place in bacteria, instead they cautiously state that it is not a major factor in translation coupling. Indeed, it is likely that some level of recycling occurs in RRF KD even after 60 minutes. The splitting could be spontaneous, assisted by backup pathways or by residual levels of RRF. Therefore, we cannot exclude a possibility that reinitiation makes some contribution to the coupling and still occur in RRF KD. Did author try to estimate what is the level of recycling in RRF KD?

2) RO differences between WT and KD in Figure 5B seems larger than what would be normally described as "little or no effect", though it is hard to understand the significance of these differences without a reference for comparison. Assuming that translation initiation should affect expression of downstream cistrons more than upstream, perhaps in addition to the analysis in Figure 6, a comparison of upstream and downstream cistrons should be carried out separately. If RRF KD has no affect on coupling we should expect the correlation between WT and KD for upstream cistrons not significantly exceed that for downstream. Further, the ratios of upstream/downstream cistrons RO vary up to 2 orders of magnitude (Figure 6B). Are they true coupled pairs within the same operons? How the same analysis would look like if done for RNA-seq reads? Would it make sense to do this with footprint densities instead of RO? Perhaps Ribosome Occupancy and Footprint Density (along with collision and 3'UTR scores for consistency) should be explicitly defined with a mathematical formula in the Materials and methods section.

3) Presumably for binning mRNAs into highly and lowly translated, RO was calculated using WT, because otherwise collisions in KD would affect RO and collisions would be overrepresented in the high bin resulting in a circular argument. This is not clear. Also, it would be interesting to see a relationship between collision scores and 3'UTR scores to support the statement about their inverse correlation. Could stacking be induced by inability of "unrecycled" ribosomes to progress to 3'UTRs?

4) The manuscript provides a very good overview of the literature related to translational coupling and reinitiation in bacteria, however, I think one relevant work is missing – Osterman et al., 2013. I think it would be good to add it to the Discussion section since it addresses many related questions and some of their observations are not in agreement with this work.

Reviewer #3:

This is a very clearly written manuscript that describes the consequences of depleting ribosome recycling factor (RRF) in *E. coli*. The authors argue that depleting RRF leads to piling up of ribosomes at the ends of ORFs, and association of non-translating ribosomes in 3' UTRs. Depleting RRF has no effect on the translational coupling of closely spaced ORFs, arguing against both of the two major models for this process. Lastly, the authors show that RRF depletion leads to induction of arfA expression, although the evidence for the proposed mechanism involving tmRNA is a little weak. The data are of high quality throughout, and the work is beautifully presented.

The authors use the piling up of ribosomes upstream of stop codons to infer that a ribosome is stalled at the stop codon in RRF-depleted cells. However, the signal from the ribosome at the stop codon is unchanged. Either the model is wrong, or ribosome profiling data cannot capture this stalled state. In the latter case, there may be other ribosome states that are not being captured, meaning that the ribosome profiling data may give a misleading view of where ribosomes are located in RRF-depleted cells. I think it's important to figure out why the stalled terminating ribosomes are not detected, including the possibility that there are no stalled terminating ribosomes.

- Figure 7F. The authors speculate that increased ribosome occupancy in tmRNA downstream of the ORF is due to translationally inactive ribosomes. This should be possible to assess using the high salt datasets.

- Figure 7F. Assuming the ORF within tmRNA is the degradation tag, the ribosome occupancy represents ribosomes translating tagged non-stop mRNAs. A simpler model for arfA induction could be that free tmRNA levels are depleted due to ribosomes getting stuck at the end of the ORF in tmRNAs rescuing non-stop mRNAs.

---

## [Author Response]

Summary:The reviewers found the work to be of high quality, and agreed that it represents an important advance in our understanding of both RRF function and translational coupling. Our one significant concern is that the bulk translation rate after RRF depletion was not measured. Interpretation of data from several experiments looking at translation coupling relies on the assumption that translation is ongoing in the RRF-depleted cells. The reviewers have a number of other suggestions that can all be addressed without the need for further experiments.Essential revisions:1) Measure the bulk translation rate across the time-course of RRF depletion. The current interpretation of Figure 3, Figure 5 and Figure 6 is based on the assumption that translation is ongoing in the RRF-depleted cells. If translation levels are greatly reduced, the conclusions about translational coupling might not be valid.

Thanks for this suggestion. We measured bulk translation rates 5, 15, and 60 minutes after the second media change (the same time points as the ribosome profiling experiments). We find roughly 40% of protein synthesis levels remain after RRF depletion, a significant level that gives us confidence in our conclusions regarding translational coupling. These data are added as Figure 1—figure supplement 1. We also note that the nanoLuc reporters in Figure 5 used to measure translational coupling are induced by addition of anhydrotetracycline at the second media change. Their transcription and translation therefore takes place under conditions of RRF depletion, further arguing that we can detect translational coupling over this time period.

2) Acknowledge the fact that residual recycling is likely occurring in the RRF-depleted cells. If it is possible to measure this residual activity, that would be great, but such an experiment is not required.

This caveat was added to the first paragraph of the Discussion section.

3) Include some discussion of the potential impact of pleiotropic effects of RRF depletion on the observed phenotypes. Given that most of the phenotypes were observed at the 5 minute time-point, the reviewers did not see this as a critical problem.

This caveat was added to the first paragraph of the Discussion section.

4) Change the title to focus solely on translational coupling, which in the opinion of the reviewers is the most important aspect of the paper.

The title was changed to focus on translational coupling as suggested.

The reviewers have additional comments, detailed below, that you should respond to with changes to the text where appropriate.Reviewer #1:Summary:Although this MS addresses the interesting question of RRF function in the translation of coupled transcripts in *E. coli*, it depends upon the correctness of two fundamental assumptions that are not examined critically in the text. The first assumption is that effects resulting from the absence of RRF protein, which is almost complete at 15' and complete at 60' (subsection “Upon RRF depletion, ribosomes accumulate in the 3’-UTR and queue upstream of stop codons”), can be solely attributed to the loss of RRF protein. Given that rrf is an essential gene, RRF removal could well have pleiotropic effects not considered by the authors. The second, and related, assumption is that ribosome occupancy is a faithful reporter of protein translation rates when RRF is absent. Missing are any direct measures of protein synthesis and cell viability that would test these assumptions.

We included the caveat about indirect effects in the Discussion section. To minimize these effects, we focus our analyses on phenomena directly related to recycling. We also note that most of these effects are seen even at the shortest time point (5 min). As described above, we directly measured protein synthesis levels over the course of the experiment to validate our use of ribosome density in the translational coupling assays.

Essential revisions:1) The present Discussion section is overlong and very speculative. It would be improved by substantial condensation. On the other hand, a critical analysis of the shortcomings of prior results on translational coupling and RRF would be valuable.

The Discussion section was edited to emphasize translational coupling. We removed two paragraphs containing speculative comments about recycling in general. With respect to prior studies with RRF, we improved the discussion of these studies in the introduction. One problem was that studies focusing on recycling did not use known translationally coupled pairs in their native context (keeping the local mRNA structure etc).

2) Subsection “Loss of RRF does not affect translational coupling in polycistronic messages” -. First paragraph. The last sentence is unclear. Also, it would be helpful to the reader to include a description of the essence of the Yamamoto et al., proposed mechanism. This might be better placed in the Discussion section.

This sentence was deleted. The mechanism proposed by Yamamoto et al., was fleshed out more in the Discussion section.

3) Figure 7F should be redrawn. At present, it is very reader unfriendly

We redrew Figure 7F so that the traces are no longer overlapping and the cartoon of the tmRNA sequence is more informative.

Reviewer #2:In this study, Saito et al. explored the role of RRF by examining how mRNA translation responses to RRF knockdown in *E. coli* using ribosome profiling and a series of ad hoc expression reporters. The major (somewhat surprising) findings are that RRF KD has relatively little effect on translational coupling and that RRF KD triggers upregulation of ArfA ribosome rescue pathway via inactivation of tmRNA by unrecycled ribosomes. This work is of an outstanding technical quality and is very important for our understanding of ribosome recycling in bacteria and of the mechanism of translational coupling.Essential revisions:1) In the absence of direct evidence, the authors are reasonably cautious not to state that translation re-initiation does not take place in bacteria, instead they cautiously state that it is not a major factor in translation coupling. Indeed, it is likely that some level of recycling occurs in RRF KD even after 60 minutes. The splitting could be spontaneous, assisted by backup pathways or by residual levels of RRF. Therefore, we cannot exclude a possibility that reinitiation makes some contribution to the coupling and still occur in RRF KD. Did author try to estimate what is the level of recycling in RRF KD?

See above in Essential revisions, point 2.

2) RO differences between WT and KD in Figure 5B seems larger than what would be normally described as "little or no effect", though it is hard to understand the significance of these differences without a reference for comparison.

We removed these data from Figure 5B.

Assuming that translation initiation should affect expression of downstream cistrons more than upstream, perhaps in addition to the analysis in Figure 6, a comparison of upstream and downstream cistrons should be carried out separately. If RRF KD has no affect on coupling we should expect the correlation between WT and KD for upstream cistrons not significantly exceed that for downstream.

As suggested, we selected pairs of genes and compared the upstream gene ribosome density in the WT and KD strains. Separately, we compared the downstream gene density in the WT and KD strains. These data have the same correlation, as expected, and consistent with our conclusions in Figure 6.

Further, the ratios of upstream/downstream cistrons RO vary up to 2 orders of magnitude (Figure 6B). Are they true coupled pairs within the same operons? How the same analysis would look like if done for RNA-seq reads?

We agree that the ratios vary more than one would expect for ORFs in the same mRNA. We were conservative in our operon definitions, using the Wanner et al., 2014 definitions of transcriptional units as curated in RegulonDB. We have now applied an additional criterion for inclusion: less than a five-fold difference in RNAseq levels between the upstream and downstream gene, because we agree that greater differences in RNA level were likely to arise from differential transcription. Figure 6 now includes these updated data but the conclusion remains the same.

Would it make sense to do this with footprint densities instead of RO? Perhaps Ribosome Occupancy and Footprint Density (along with collision and 3'UTR scores for consistency) should be explicitly defined with a mathematical formula in Materials and methods section.

The data shown in Figure 6B are ratios of footprint densities, not RO. We have included more description of these metrics in the Materials and methods section.

3) Presumably for binning mRNAs into highly and lowly translated, RO was calculated using WT, because otherwise collisions in KD would affect RO and collisions would be overrepresented in the high bin resulting in a circular argument. This is not clear. Also, it would be interesting to see a relationship between collision scores and 3'UTR scores to support the statement about their inverse correlation. Could stacking be induced by inability of "unrecycled" ribosomes to progress to 3'UTRs?

mRNAs were binned using RO values from the WT sample, as you say, and then the density from the KD is shown in Figure 4. The text and caption were altered to make this clear. Regarding the last point, we agree that stacking is caused by post-TCs stalled at stop codons. But whether or not the post-TC remains there or slides into the 3’-UTR depends on the ribosome occupancy of the gene (which is at least in part determined by its initiation rate). The simplest way to explain the correlation between RO and stacking (and lower 3’-UTR density) is to postulate that stacking prevents post-TCs from progressing into the 3’-UTR.

4) The manuscript provides a very good overview of the literature related to translational coupling and reinitiation in bacteria, however, I think one relevant work is missing – Osterman et al., 2013. I think it would be good to add it to the Discussion section since it addresses many related questions and some of their observations are not in agreement with this work.

We have decided not to include this reference because although it describes experiments aimed at understanding the mechanism of translational coupling, it does not specifically address recycling or RRF. Like many papers on coupling, it cannot distinguish between models of re-initiation and the effects of RNA structure on de novo initiation.

Reviewer #3:This is a very clearly written manuscript that describes the consequences of depleting ribosome recycling factor (RRF) in *E. coli*. The authors argue that depleting RRF leads to piling up of ribosomes at the ends of ORFs, and association of non-translating ribosomes in 3' UTRs. Depleting RRF has no effect on the translational coupling of closely spaced ORFs, arguing against both of the two major models for this process. Lastly, the authors show that RRF depletion leads to induction of arfA expression, although the evidence for the proposed mechanism involving tmRNA is a little weak. The data are of high quality throughout, and the work is beautifully presented.The authors use the piling up of ribosomes upstream of stop codons to infer that a ribosome is stalled at the stop codon in RRF-depleted cells. However, the signal from the ribosome at the stop codon is unchanged. Either the model is wrong, or ribosome profiling data cannot capture this stalled state. In the latter case, there may be other ribosome states that are not being captured, meaning that the ribosome profiling data may give a misleading view of where ribosomes are located in RRF-depleted cells. I think it's important to figure out why the stalled terminating ribosomes are not detected, including the possibility that there are no stalled terminating ribosomes.

We recognize that it is unsatisfying that we cannot directly observe a strong increase in post-TCs at stop codons. We looked for stop codon peaks in reads of all sizes, thinking that perhaps these reads were shorter or longer than normal, but did not find an increase in stop peaks with any footprint size. It may be that collisions with upstream elongating ribosomes prevent us from isolating the relevant ribosome footprints. Nevertheless, we are confident that the queuing of elongating ribosomes upstream of stop codons is strong indirect evidence of post-TCs stalled at stop codons.

- Figure 7F. The authors speculate that increased ribosome occupancy in tmRNA downstream of the ORF is due to translationally inactive ribosomes. This should be possible to assess using the high salt datasets.

This is a good suggestion. We observe fewer ribosome footprints on tmRNA in the high salt datasets. A sentence was added to this effect in the Results section.

- Figure 7F. Assuming the ORF within tmRNA is the degradation tag, the ribosome occupancy represents ribosomes translating tagged non-stop mRNAs. A simpler model for arfA induction could be that free tmRNA levels are depleted due to ribosomes getting stuck at the end of the ORF in tmRNAs rescuing non-stop mRNAs.

Yes, this is our model, that ribosomes stuck on tmRNA inhibit its activity, leading to activation of ArfA.